# Vitamin D Attenuates Fibrotic Properties of Fibrous Dysplasia-Derived Cells for the Transit towards Osteocytic Phenotype

**DOI:** 10.3390/ijms25094954

**Published:** 2024-05-01

**Authors:** Ha-Young Kim, Jung-Hee Shim, Baek-Kyu Kim, Chan-Yeong Heo

**Affiliations:** 1Interdisciplinary Program in Bioengineering, Seoul National University, Seoul 08826, Republic of Korea; hkim247@snu.ac.kr; 2Department of Plastic and Reconstructive Surgery, College of Medicine, Seoul National University, Seoul 03080, Republic of Korea; plasrecon@snubh.org; 3Department of Research Administration Team, Seoul National University Bundang Hospital, Seongnam 13620, Republic of Korea; 98757@snubh.org; 4Department of Plastic and Reconstructive Surgery, Seoul National University Bundang Hospital, Seongnam 13620, Republic of Korea

**Keywords:** fibrous dysplasia, bone disease, vitamin D, fibrosis, mineralization

## Abstract

Fibrous dysplasia (FD) poses a therapeutic challenge due to the dysregulated extracellular matrix (ECM) accumulation within affected bone tissues. In this study, we investigate the therapeutic potential of 1,25-dihydroxyvitamin D3 (1,25(OH)_2_D_3_) in managing FD by examining its effects on FD-derived cells in vitro. Our findings demonstrate that 1,25(OH)_2_D_3_ treatment attenuates the pro-fibrotic phenotype of FD-derived cells by suppressing the expression of key pro-fibrotic markers and inhibiting cell proliferation and migration. Moreover, 1,25(OH)_2_D_3_ enhances mineralization by attenuating pre-osteoblastic cellular hyperactivity and promoting maturation towards an osteocytic phenotype. These results offer valuable insights into potential treatments for FD, highlighting the role of 1,25(OH)_2_D_3_ in modulating the pathological properties of FD-derived cells.

## 1. Introduction

Fibrous dysplasia (FD) represents a complex interplay between aberrant cellular activities and dysregulated extracellular matrix (ECM) dynamics, culminating in the accumulation of dense fibrous tissue within affected bone regions. The pathogenesis of FD involves atypical pre-osteoblastic and mesenchymal stromal fibroblastic cell populations, whose actions drive the pathological alterations observed in FD lesions [1,2]. Central to these aberrations is the *GNAS* mutation in bone marrow mesenchymal stem cells (BMSCs), which disrupt cyclic AMP (cAMP)-associated pathways to trigger a cascade of pathological events [3].

At the forefront of FD pathogenesis lies the perturbation of osteogenic differentiation, resulting in the entrapment of stromal cells in a pre-osteoblastic state. The maturation of pre-osteoblasts, pivotal for normal bone formation, is arrested in FD, leading to an accumulation of undifferentiated cells with dysregulated activities [4]. These cells express pro-fibrotic factors like transforming growth factor beta (TGFβ), collagen type 1 alpha 1 chain (COL1A1), collagen type 3 alpha 1 chain (COL3A1), procollagen lysine 2-oxoglutarate 5-dioxygenase (PLOD), and periostin (POSTN), which contribute to aberrant ECM production and deposition for the formation of dense fibrotic tissue [5,6,7].

Moreover, the *GNAS* mutation exerts its influence beyond osteogenic differentiation, potentially altering the differentiation or de-differentiation of other cell types into fibroblasts [8,9], further propagating fibrous tissue accumulation. The ensuing overproduction of cAMP, a hallmark of the *GNAS* mutation, intricately modulates pro-fibrotic responses through various mechanisms. Elevated cAMP levels stimulate fibrotic proliferation and ECM synthesis while promoting the expression of pro-fibrotic factors such as TGFβ [10]. The TGFβ pathway orchestrates the upregulation of pro-fibrotic genes such as *COL1A1*, *COL3A1*, connective tissue growth factor (*CTGF*), and fibronectin (*FBN*) for the cellular transition from a quiescent to an activated state [11,12,13]. Upon activation, fibroblasts undergo phenotypic alterations characterized by heightened proliferative capacity, increased migratory potential, and augmented ECM synthesis [14], substantiating the development of tissue stiffness and fibrous tissue formation.

Despite advancements in the understanding of FD pathophysiology, therapeutic interventions remain largely elusive and limited to symptomatic control. Current investigatory drugs include anti-bone resorptive agents such as bisphosphonates [15], denosumab [16], and tocilizumab [17], aimed at attenuating the accelerated bone turnover, alongside analgesics like tanezumab [18] and anti-brain-derived neurotrophic factor (BDNF) [19] for alleviating associated bone pain. Surgical interventions may also be employed to address deformities or fractures resulting from the disease process. However, it is important to note that these treatments are merely palliative and fail to target the fundamental molecular and cellular aberrations involving mineralization and fibrosis that drive FD pathogenesis. Consequently, patients are left with limited viable options for disease management or treatment.

In light of these limitations, vitamin D has emerged as a potential therapeutic candidate for FD management. Known for its roles in bone metabolism, mineralization, and its anti-proliferative/anti-inflammatory properties, vitamin D enhances calcium absorption and promotes osteoblast differentiation and function [20] to counter the dysregulated turnover observed in FD lesions and facilitate the formation of structurally sound bone tissue. Evidence also suggests it has anti-proliferative effects on mesenchymal stem cells [21], crucial in FD pathogenesis, and potential to curb pro-fibrotic phenotypes in diseases such as lung and liver fibrosis. By inhibiting abnormal cell proliferation, vitamin D supplementation could regulate the expansion of fibroblastic and immature osteogenic cell populations within FD lesions, thereby attenuating disease progression.

Recognizing the potential of vitamin D, our study investigated the impact of 1,25-dihydroxyvitamin D_3_ (1,25(OH)_2_D_3_), an active form of vitamin D_3_, on mitigating the pro-fibrotic phenotype and promoting the maturation and mineralization of FD-derived cells in vitro. We aimed to elucidate the therapeutic potential of 1,25(OH)_2_D_3_ by exploring the cellular responses of FD-derived cells to its treatment, focusing on its effects on migrative and proliferative properties, pre-osteoblastic hyperactivity, and osteogenic maturation/mineralization, in comparison to normal BMSCs. Through our investigation, we sought to provide insights into the benefits of 1,25(OH)_2_D_3_ in addressing aberrant ECM development associated with FD.

## 2. Materials and Methods

### 2.1. Human Primary Tissues

Fresh tissue samples were collected from five patients diagnosed with craniofacial fibrous dysplasia undergoing surgery for lesion resections and normal craniofacial bone samples from healthy volunteers undergoing cosmetic facial bone contouring surgery. The collection and use of surgical specimens for research were carried out in accordance with approved protocols by the Institutional Review Board of Seoul National University Bundang Hospital (B-2111-718-302) and Samsung Medical Center (2021-12-025). The demographic and clinical features of the donors are summarized in Table 1.

### 2.2. Specimen Dissociation and Cell Isolation

Fresh lesion specimens were minced into fine particles and then subjected to digestion with a 2 mg/mL collagenase D solution (Roche, Mannheim, Germany) in growth medium (Dulbecco’s Modified Eagle medium (DMEM; Gibco Life Technologies, Carlsbad, CA, USA) supplemented with 10% fetal bovine serum (FBS) and 1% antibiotic–antimycotic solution). This digestion process lasted for 4 h at 37 °C with gentle agitation. The resulting cell suspension was filtered through a 70 μm cell strainer, treated with 1x RBC Lysis Buffer (Invitrogen, San Diego, CA, USA), and either cryopreserved or cultured in T-flasks at a density of 3 × 10^5^ cells per flask. Any undigested residues were maintained in growth medium for primary explant culture. Approximately two weeks post-explantation, cells were collected and subjected to cryopreservation or replating in fresh culture dishes.

Normal bone specimens from healthy volunteers, devoid of associated metabolic bone pathology, were used to isolate bone marrow mesenchymal stromal cells (BMSCs) for the control group. The marrow was scraped into basal medium, followed by pipetting and serial passages through needles of decreasing diameter. Cells were either cryopreserved or replated for culture.

### 2.3. RNA Extraction and RT-qPCR

Cells were seeded into 6-well plates at a density of 2 × 10^5^ cells per well and allowed to attach to the plate for 24 h before treating with either 100 nM of 1,25(OH)_2_D_3_ (Sigma–Aldrich, Waltham, MA, USA) or 10 nM of prostaglandin E2 (PGE2) (Sigma-Aldrich) or a combination of both in growth medium. After 24 h, RNA extraction was performed, and subsequently, cDNA was synthesized utilizing the cDNA Synthesis Kit (Thermo Scientific, Waltham, MA, USA). The mRNA expressions were quantified via real-time PCR using Power SYBR Green^®^ PCR Master Mix on a QuantStudio™ 7 Flex PCR System (Applied Biosystems, Waltham, MA, USA). Primers used in the analysis are listed in Table 2.

### 2.4. Wound Healing Assay

Cells were seeded at 5 × 10^4^ cells per well into 12-well plates and allowed to reach confluence. A 200 μL pipette tip was used to create a uniform scratch across the cell monolayer. Cells were then washed with phosphate-buffered saline (PBS) and treated with either 100 nM of 1,25(OH)_2_D_3_ or 10 nM of PGE2 or a combination of both in growth medium. Images were captured at 0 and 16 h post-wounding using an inverted microscope. The migration rate was determined by measuring the wound area using ImageJ software (version 1.54i). Migration was expressed as the percentage of wound closure relative to the initial wound area.

### 2.5. Proliferation Assay

Cells were seeded into 96-well plates at a density of 5 × 10^3^ cells per well and allowed to attach to the plate for 24 h before treating with either 100 nM of 1,25(OH)_2_D_3_ or 10 nM of PGE2 or a combination of both in growth medium. After 24 h, the proliferation rate was assessed using the 3-(4,5-dimethylthiazol-2-yl)-2,5-diphenyltetrazolium bromide (MTT) solution (Invitrogen). Briefly, cells were incubated with MTT solution (0.5 mg/mL) for 4 h at 37 °C. The formazan crystals formed were solubilized with dimethyl sulfoxide (DMSO), and the absorbance was measured at 570 nm using a microplate reader.

### 2.6. Immunofluorescence

Cells were seeded at 5 × 10^4^ cells per well into 12-well plates and allowed to attach to the plate for 24 h before treating with either 100 nM of 1,25(OH)_2_D_3_ or 10 nM of PGE2 or a combination of both in growth medium. After 48 h, cells were fixed with 4% paraformaldehyde for 10 min at RT, permeabilized with 0.5% Triton X-100, blocked with UltraCruz^®^ blocking reagent (Santa Cruz Biotech, Santa Cruz, CA, USA) for 30 min at RT, and incubated with primary antibodies overnight at 4 °C with appropriate primary antibodies: anti-COL1, anti-COL3, and anti-TGFβ1 (all from Santa Cruz, diluted at 1:100 in blocking reagent). The following day, Alexa Fluor^®^-labeled secondary antibody (Invitrogen, diluted at 1:250 in blocking reagent) was applied for 1 h at RT before mounting in Vectashield mounting medium with DAPI (VMR). Final histological images were taken using Zeiss LSM800 confocal microscope (Zeiss, Oberkochen, Germany) after curing the mounting medium overnight.

### 2.7. ALP and ARS Assay

For the alkaline phosphatase (ALP) assay, cells were seeded into 96-well plates at a density of 5 × 10^3^ cell per well and allowed to reach 60% confluence in the growth medium prior to treating with either 100 nM of 1,25(OH)_2_D_3_ or 10 nM of PGE2 in osteogenic differentiation medium (DMEM supplemented with 10% FBS, 10 nM dexamethasone, 50 μg/mL ascorbic acid, 10 mM sodium β-glycerophosphate, and 1% antibiotic-antimycotic). A fresh medium change with treatment was executed every 3 days for 2 weeks. Cells were fixed with 4% paraformaldehyde for 10 min at RT, and subsequently incubated with an ALP staining solution (Takara Bio, Tokyo, Japan) for 30 min. Cells were imaged using a light microscope. For quantitative analysis, the bound dye was eluted from stained cells using DMSO, and the absorbance of the eluted dye was measured using a spectrophotometer.

For the Alizarin Red S (ARS) assay, cells were cultured and treated in the same manner as outlined previously for the ALP assay. However, osteogenic differentiation was induced in cells for a longer period of time of 3 weeks. Cells were fixed with 4% paraformaldehyde for 10 min at RT and incubated with 2% ARS solution (VWR International, Radnor, PA, USA) for 1 h. Stained cells were visualized under a light microscope, and images were captured to assess calcium deposition as red-stained mineralized nodules. For quantitative analysis, the bound dye was eluted from stained cells using DMSO, and the absorbance of the eluted dye was measured using a spectrophotometer.

### 2.8. Statistical Analysis

Statistical analysis and data representation were performed using GraphPad Prism 8. The statistical differences were determined by one-way ANOVA. All quantitative results are presented as the mean ± standard deviation of five independent experiments, with each performed in triplet, if not noted otherwise. A *p*-value < 0.05 was considered statistically significant.

## 3. Results

### 3.1. 1,25(OH)_2_D_3_ Suppresses Pro-Fibrotic Properties of FD-Derived Cells

A histological analysis of FD-derived cells revealed the significantly elevated expression of key pro-fibrotic markers—COL1, COL3, and TGFβ1—in comparison to normal BMSCs (Figure 1A). Upon a low dose of stimulation with PGE2, these markers exhibited a robust increase in expression, which was effectively attenuated by 1,25(OH)_2_D_3_ treatment. These differences were more prominent in FD-derived cells, which displayed strong fibrotic phenotypes compared to normal BMSCs. Our analysis of the mRNA expression levels corroborated these histological observations, demonstrating a significant suppression in the expression profile of *COL1A1*, *COL3A1*, and *TGFβ1* in the presence of 1,25(OH)_2_D_3_ (Figure 1B).

To investigate the modulatory effects of 1,25(OH)_2_D_3_ on the pro-fibrotic phenotypes, we examined its impact on the migratory and proliferative capacities of FD-derived cells. Migration assays revealed a heightened migratory propensity of FD-derived cells, evidenced by their increased migration distances and rates relative to normal BMSCs (Figure 1C). Treatment with 1,25(OH)_2_D_3_ significantly curtailed the migratory capacity of FD-derived cells, both in the presence and absence of PGE2 stimulation. These findings suggest that 1,25(OH)_2_D_3_ suppresses the migratory attributes of FD-derived cells, potentially inhibiting their tissue infiltration ability. Furthermore, FD-derived cells exhibited an augmented proliferative capacity compared to normal BMSCs, which was significantly attenuated with the treatment of 1,25(OH)_2_D_3_, particularly in PGE2-stimulated, FD-derived cells (Figure 1D). Collectively, these findings delineate the efficacy of 1,25(OH)_2_D_3_ in mitigating the proliferation and migration of FD-derived cells, thus highlighting the therapeutic potential in combating pro-fibrotic phenotypes associated with FD.

### 3.2. 1,25(OH)_2_D_3_ Promotes Osteoblast Maturation and Enhances Mineralization

Treatment with 1,25(OH)_2_D_3_ demonstrated a notable impact on the hyperactivity and mineralizing capacities of FD-derived cells. Firstly, we evaluated the effect of 1,25(OH)_2_D_3_ on the hyperactivity of FD-derived cells using the ALP assay (Figure 2A). FD-derived cells treated with 1,25(OH)_2_D_3_ exhibited a significant attenuation of elevated ALP activity compared to untreated cells, indicative of suppressed hyperactivity. This reduction in ALP activity suggests that 1,25(OH)_2_D_3_ treatment effectively mitigates the hyperactivity of FD-derived cells, potentially normalizing their osteogenic differentiation process.

Furthermore, we investigated the mineralizing capacities of FD-derived cells in response to 1,25(OH)_2_D_3_ treatment using the ARS assay. Treatment with 1,25(OH)_2_D_3_ significantly enhanced mineralization in FD-derived cells compared to untreated cells, which demonstrated significantly low levels of ARS activity, as evidenced by the increased Alizarin Red S staining intensity. This enhancement in mineralizing capacities suggests that 1,25(OH)_2_D_3_ treatment promotes the formation of mineralized bone tissue via FD-derived cells, potentially counteracting the dysregulated bone remodeling observed in fibrous dysplasia. Overall, our results demonstrate that 1,25(OH)_2_D_3_ treatment effectively mitigates hyperactivity and enhances the mineralizing capacities of FD-derived cells.

The mRNA expression levels of early- to late-osteogenic markers were assessed to investigate the maturation status of FD-derived cells following treatment with 1,25(OH)_2_D_3_ (Figure 2B). Our results revealed the inhibition of Runt-related transcription factor 2 (*RUNX2*), an early osteogenic marker, which was highly expressed in FD-derived cells compared to normal BMSCs. Furthermore, the suppressed expression of late-osteogenic markers, including osteocalcin (*OCN*), dentin matrix protein 1 (*DMP1*) and sclerostin (*SOST*), was restored in FD-derived cells treated with 1,25(OH)_2_D_3_ compared to untreated cells, aligning them with the expression levels observed in normal BSMCs. OCN, DMP1, and SOST are key components of the ECM in bone tissue and are synthesized by mature osteoblasts during the late stages of osteogenesis. Therefore, their increased expression indicates a shift towards a mature osteoblast/osteocyte phenotype upon treatment with 1,25(OH)_2_D_3_. These findings suggest that 1,25(OH)_2_D_3_ promotes the differentiation and maturation of FD-derived cells towards a bone-forming phenotype, which may contribute to the restoration of normal bone remodeling and mineralization in FD.

## 4. Discussion

Fibrous dysplasia (FD) presents a challenging clinical scenario, characterized by the pathological replacement of normal bone with fibrous tissue. This disorder manifests with bone deformities, fractures, and chronic pain [22], often necessitating therapeutic interventions to mitigate its progression and associated symptoms. The pathogenesis of FD involves the intricate dysregulation of bone ECM formation, orchestrated by atypical mesenchymal stromal fibroblastic and pre-osteoblastic cell populations within FD lesions [23]. In this study, we sought to explore the therapeutic potential of 1,25(OH)_2_D_3_ in addressing the fibrotic phenotypes and aberrant bone metabolism characteristic of FD (Figure 3). Our results demonstrate that treatment with 1,25(OH)_2_D_3_ effectively mitigates the pro-fibrotic properties of FD-derived cells. We observed a significant reduction in the mRNA levels of fibrotic markers, including *COL1A1*, *COL3A1*, and *TGFβ1*, in FD-derived cells treated with 1,25(OH)_2_D_3_. These findings suggest that 1,25(OH)_2_D_3_ suppresses the fibrotic phenotype of FD-derived cells, potentially inhibiting the abnormal accumulation of fibrous tissue within FD lesions.

Furthermore, we found that treatment with 1,25(OH)_2_D_3_ significantly reduces the migratory and proliferative properties of FD-derived cells. This observation highlights the anti-migratory and anti-proliferative effects of 1,25(OH)_2_D_3_, which may contribute to its therapeutic potential in mitigating the progression of FD. Additionally, we demonstrated that 1,25(OH)_2_D_3_ treatment enhances the mineralizing capacities of FD-derived cells, as evidenced by increased ALP activity and ARS staining intensity. These findings suggest that 1,25(OH)_2_D_3_ promotes osteogenic differentiation and mineralization in FD-derived cells, potentially restoring normal bone metabolism and structure in FD lesions. The therapeutic effects of 1,25(OH)_2_D_3_ observed in our study are supported by previous studies demonstrating its anti-fibrotic [24,25], anti-proliferative [26,27], and pro-osteogenic [28] properties in various disease models. Vitamin D_3_ has been shown to regulate key signaling pathways involved in fibrosis, proliferation, and osteogenesis, including the TGFβ1/mothers against decapentaplegic homolog (Smad) and wingless-type (Wnt)/β-catenin pathways [29,30]. By modulating these pathways, 1,25(OH)_2_D_3_ may exert its therapeutic effects by inhibiting fibrotic processes, suppressing abnormal cell proliferation, and promoting osteogenic differentiation.

While traditionally fibroblasts were not considered direct precursors of osteoblasts, and vice versa, emerging evidence suggests that they possess differentiation plasticity, allowing for transdifferentiation processes [31]. This phenomenon is evident in certain pathological contexts like ossification of the posterior longitudinal ligament (OPLL), where fibroblasts within ligament tissue differentiate into osteoblast-like cells, contributing to ectopic bone formation [32,33]. Similarly, in conditions such as X-linked hypomyelination with spondylometaphyseal dysplasia (H-SMD), patient-derived fibroblasts have demonstrated the ability to transdifferentiate into osteoblast-like cells to recapitulate the disease-relevant skeletal phenotype [34]. Although their relevance to FD is speculative, the shared fibroblastic phenotype expressed across various cellular subsets within FD lesions suggest the potential for interconversion between fibroblasts and other differentiated progenies of MSC [23].

The significantly greater amounts of alizarin Red S content, indicative of increased calcium deposition, and reduced fibrogenic marker expressions in FD-derived cells treated with 1,25(OH)_2_D_3_, support the shift towards osteoblastic differentiation. This aligns with the notion that stromal fibroblasts within FD lesions may exhibit osteoblastic phenotypic features under environmental stimuli, such as vitamin D-associated signaling, further highlighting the potential for cellular plasticity modulation and transdifferentiation processes in FD pathogenesis. However, the specific mechanisms underlying the interplay between fibroblasts and osteoblasts, as well as their potential transdifferentiation into osteoblastic cells, remain unclear in FD. Further research is needed to elucidate the cellular dynamics underlying FD and to investigate the precise mechanisms through which vitamin D_3_ exerts its effects on cellular differentiation processes, offering a novel approach for the management of FD.

This study provided valuable insights into how 1,25(OH)_2_D_3_ can effectively intervene in the multifaceted pathology of FD. By demonstrating its ability to mitigate the pro-fibrotic properties of FD-derived cells while simultaneously promoting osteogenic differentiation and mineralization, our findings underscore the therapeutic potential of 1,25(OH)_2_D_3_ as a comprehensive treatment that targets both fibrotic phenotypes and aberrant bone metabolism in FD. Importantly, while investigatory drug candidates such as the ones mentioned earlier led to insignificant radiographic improvements and complications like hypercalcemia [35] or nausea and myalgia [36], this vitamin D analog offers a potent and safe alternative.

However, an important limitation of this study lies in the lack of investigation into specific signaling pathways mediating these effects. Vitamin D_3_ acts through the vitamin D receptor (VDR) to modulate various signaling cascades, including the Wnt/β-catenin pathway, mitogen-activated protein kinase (MAPK) pathway, and phosphatidylinositol 3-kinase (PI3K)/(protein kinase B) Akt pathway [37]. The activation of these pathways can influence cell proliferation, differentiation, and survival, all of which are relevant to FD pathogenesis. By elucidating the downstream signaling pathways affected by 1,25(OH)_2_D_3_ in FD-derived cells, we could gain a deeper understanding of its mechanisms of action and potentially identify novel therapeutic targets. Additionally, pathway analysis could provide insights into the crosstalk between vitamin D_3_ signaling and other pathways implicated in FD, such as the cAMP pathway associated with the *GNAS* mutation.

Therefore, future studies should consider integrating pathway analysis techniques, such as transcriptomic profiling or phosphoproteomic analysis, to elucidate the intricate molecular mechanisms underlying the therapeutic effects of vitamin D_3_ in FD. This comprehensive approach would not only enhance our understanding of FD pathophysiology but also facilitate the development of targeted and personalized therapeutic strategies for this challenging disorder. In conclusion, our findings suggest that 1,25(OH)_2_D_3_ holds promise as a therapeutic agent for the management of fibrous dysplasia. By targeting fibrotic phenotypes and aberrant bone metabolism, 1,25(OH)_2_D_3_ may offer a novel approach for the treatment of FD. Further studies are warranted to elucidate the underlying mechanisms of action of 1,25(OH)_2_D_3_ and to evaluate its efficacy and safety in preclinical and clinical settings.

## Figures and Tables

**Figure 1 ijms-25-04954-f001:**
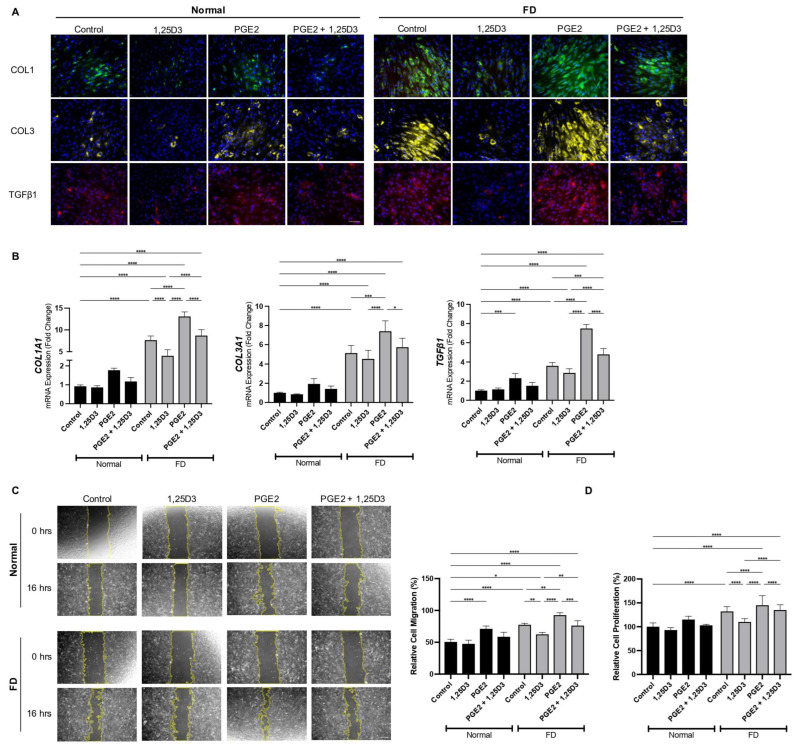
1,25(OH)_2_D_3_ inhibits the expression of fibrotic markers and attenuates cell migration and proliferation. (**A**) Representative immunofluorescent images and (**B**) mRNA expression levels of fibrotic markers COL1, COL3, and TGF-β1 under the treatment of 1,25(OH)_2_D_3_, PGE2, or both, in normal BMSCs and patient-derived cells. Functional studies to determine their effects on (**C**) cell migration and (**D**) cell proliferation were performed. The graphs represent the gene expression relative to the normal BMSCs control group. Asterisks indicate statistical significance based on one-way ANOVA (**** *p* value < 0.0001, *** *p* value < 0.001, ** *p* value < 0.01, * *p* value < 0.05). Scale bar, 100 μm.

**Figure 2 ijms-25-04954-f002:**
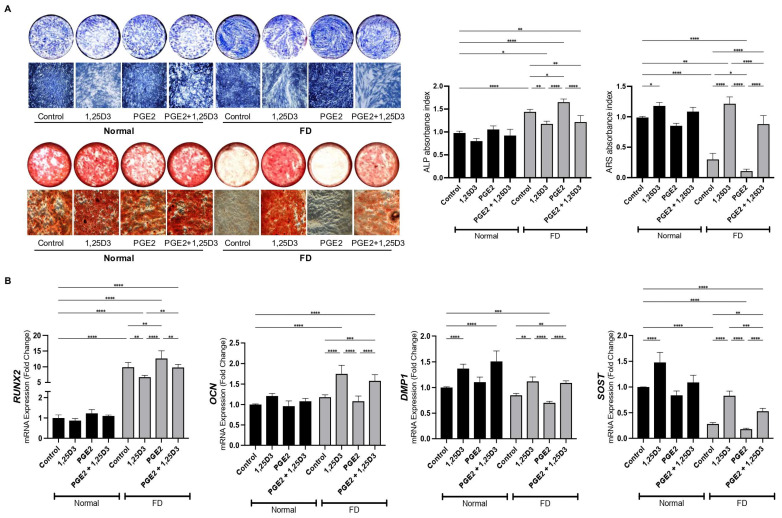
1,25(OH)_2_D_3_ enhances osteogenic maturation and mineralizing ability of FD-derived cells. (**A**) Representative images and quantitative analysis of ALP and ARS staining. (**B**) mRNA expression levels of early- to late-stage osteogenic markers. Asterisks indicate statistical significance based on one-way ANOVA (**** *p* value < 0.0001, *** *p* value < 0.001, ** *p* value < 0.01, * *p* value < 0.05).

**Figure 3 ijms-25-04954-f003:**
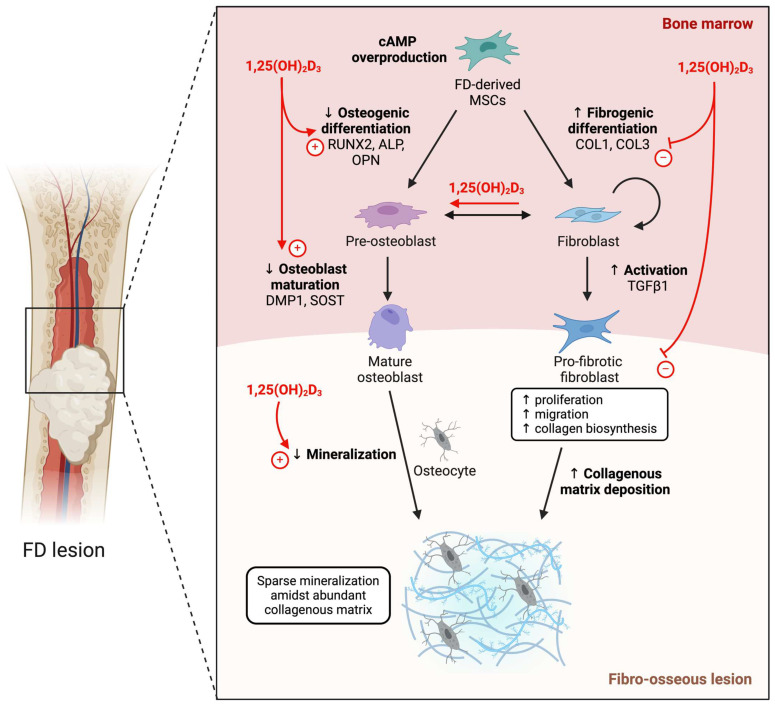
Proposed effect of 1,25(OH)_2_D_3_ on the pathobiological features of FD. 1,25(OH)_2_D_3_ suppresses the heightened release of pro-fibrogenic mediators and promotes the release of osteogenic mediators by MSCs in FD individuals, thereby impeding fibrogenic differentiation and activation into pro-fibrotic fibroblasts while fostering osteogenic differentiation and maturation to restore aberrant matrix deposition and impaired mineralization. FD: fibrous dysplasia; MSC: mesenchymal stem cell; RUNX2: Runt-related transcription factor 2; ALP: alkaline phosphatase; OPN: osteopontin; DMP1: dentin matrix protein 1; SOST: sclerostin; COL1: collagen type 1; COL3: collagen type 3; TGFβ1: transforming growth factor beta 1.

**Table 1 ijms-25-04954-t001:** Donor characteristics and clinical features.

Diagnosis	Donor	Gender	Age	Site
Fibrous Dysplasia	R1	M	18	Nasal cavity/maxillary sinus
Fibrous Dysplasia	R2	M	19	Zygomaticomaxillary
Fibrous Dysplasia	R3	F	25	Hemiface/mandible
Fibrous Dysplasia	R4	M	12	Mandible
Fibrous Dysplasia	R5	M	14	Forehead/upper orbit
None	H1	F	28	Zygomatic/mandible
None	H2	F	23	Hemiface/mandible
None	H3	M	22	Zygomatic/mandible
None	H4	M	19	Zygomatic/mandible
None	H5	F	25	Zygomatic/hemiface

R = fibrous dysplasia patient, H = healthy volunteer.

**Table 2 ijms-25-04954-t002:** Target genes and their primer sequences.

Gene Name	Forward Primer (5′-3′)	Reverse Primer (5′-3′)
*COL1A1*	GTGCGATGACGTGATCTGTGA	CGGTGGTTTCTTGGTCGGT
*COL3A1*	TGGTCTGCAAGGAATGCCTGGA	TCTTTCCCTGGGACACCATCAG
*TGFβ1*	TCGCCAGAGTGGTTATCTT	TAGTGAACCCGTTGATGTCC
*RUNX2*	TGGTTACTGTCATGGCGGGTA	TCTCAGATCGTTGAACCTTGCTA
*OCN*	CACTCCTCGCCCTATTGGC	CCCTCCTGCTTGGACACAAAG
*DMP1*	GATCAGCATCCTGCTCATGTT	AGCCAAATGACCCTTCCATTC
*SOST*	CCCTTTGAGACCAAAGACGTG	GGCCCATCGGTCACGTAG
GAPDH	ACAGTTGCCATGTAGACC	TTTTTGGTTGAGCACAGG

## Data Availability

The data presented in this study are available in this article.

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
