# Peer review of "Vitamin D Attenuates Fibrotic Properties of Fibrous Dysplasia-Derived Cells for the Transit towards Osteocytic Phenotype"

_ijms, 2024, doi:10.3390/ijms25094954_

Round 1

Reviewer 1 Report

Comments and Suggestions for Authors

Dear authors, the article entitled 'Vitamin D Attenuates Fibrotic Properties of Fibrous Dysplasia Derived Cells for the Transit Towards Osteocytic Phenotype' is well-prepared. The reseach design in approproate. The abstract, rersult and discussion are well-written. I recommend acceptance for publication to this paper after the author improve the article as follows:
1. Introduction: Please refer the previous published researches within similar area of studies. Explain their limitations, and thus you can show the novelty of this research
2. Figures: Please make it more readable by adjusting size etc. 

Author Response

Dear authors, the article entitled 'Vitamin D Attenuates Fibrotic Properties of Fibrous Dysplasia Derived Cells for the Transit Towards Osteocytic Phenotype' is well-prepared. The research design in appropriate. The abstract, result and discussion are well-written.

Author response: Thank you for your positive feedback. We are pleased to hear that you found our manuscript to be well-designed and well-written. Your time and effort dedicated to reviewing our work are greatly appreciated.

I recommend acceptance for publication to this paper after the author improve the article as follows:

1. Introduction: Please refer the previous published researches within similar area of studies. Explain their limitations, and thus you can show the novelty of this research.

Author response: In response to your comment, we have included research on investigative therapeutic interventions for FD and discussed their limitations to emphasize the novelty and relevance of our investigation into vitamin D in the introduction as well as the discussion section.

2. Figures: Please make it more readable by adjusting size etc.

Author response: Thank you for your feedback. We have resized all images and made adjustments to the fonts accordingly. We hope these changes adequately address your concern about the readability of the figures.

Reviewer 2 Report

Comments and Suggestions for Authors

This study investigated the impact of 1,25-dihydroxyvitamin D3 (1,25(OH)2D3), an 

active form of vitamin D3, on mitigating the pro-fibrotic phenotype and promoting the 

maturation and mineralization of FD-derived cells in vitro. The authors aimed to elucidate the therapeutic potential of 1,25(OH)2D3 by exploring the cellular responses of FD-derived cells to its treatment, focusing on its effects on migrative and proliferative properties, preosteoblastic hyperactivity, and osteogenic maturation/mineralization, in comparison to normal BMSCs.

The introduction is well written , with adequate bibliographic references . The objective of the study is  clearly established. It might be interesting to add a brief comment about the therapeutic options in FD. The methodology is complete, widely described, which would allow the study to be carried out by another research group. It should be noted that the control group does not present associated metabolic bone pathology The results are clear expressed  and easy to understand The discussion is adapted to the results obtained. The authors  should express the limitations and strengths of the study. A graphic representation of the mechanisms of action of the vitamin D on these cells could be interesting

Author Response

We sincerely appreciate your thorough evaluation of our study and your positive feedback on the manuscript’s design and clarity. Taking your suggestion into account, we have included information on current therapeutic options for FD and discussed their limitations to emphasize the novelty and relevance of our investigation into vitamin D in the introduction section. Furthermore, we made note of the absence of associated metabolic bone pathology in the control group. Your suggestion regarding a graphic representation of the mechanisms of action of vitamin D on FD-derived cells was very insightful, and we have incorporated Figure 3 to enhance the understanding of our findings. Additionally, we have expanded the discussion section to highlight the novelty and strengths of our study, aiming to address your comment on the study’s limitations and strengths. Thank you again for your valuable feedback.

Reviewer 3 Report

Comments and Suggestions for Authors

The subject of this study is very interesting and novel (therapeutic potential of 1,25-dihydroxyvitamin D3 (1,25(OH)2D3) in vitro in fibrous dysplasia).

The novelty of the study is supported by the fact that I have only found 9 articles in pubmed in the last 5 years relating vitamin D and fibrous dysplasia.

The introduction is well elaborated and well documented, which allows the authors to focus on the topic chosen by the authors..

The methodology is very well explained, which could allow replication of the study if the steps taken by these authors are followed.

The degree of specificity in this section is to be appreciated, given the complexity of the subject, which makes it easy to understand even for people who are not very familiar with the subject.

The results are very well adapted to the objectives of the research and are very well supported by the various figures that also allow an easy understanding of the results.

The discussion, although well structured, is perhaps excessively short, lacking a paragraph to further reinforce the results obtained, which are very notable.

It would also be advisable to include a paragraph to highlight the strengths and limitations of the study.

The bibliography is adequate but perhaps somewhat obsolete for such a new topic. The obsolescence index (median age of the references) is 10 years, only 25% are less than 5 years old, 16.7% are less than 3 years old and only $8.3 is less than 1 year old.

Author Response

Thank you for your thorough and insightful evaluation of our study. Your dedication to reviewing our manuscript is greatly appreciated. We are delighted to hear that you found our research to be interesting and novel.

In response to your feedback, we have expanded the discussion section to highlight the novelty and therapeutic implications of our findings and outline the strengths and limitations of the study.

With the added contents, references have been further supplemented as well as updated to include more recent publications.

Thank you once again for your valuable input, which contributed significantly to the improvement of our manuscript.